# Feasibility of Trastuzumab-Deruxtecan in the Treatment of Ovarian Cancer: A Systematic Review

**DOI:** 10.3390/jcm14238483

**Published:** 2025-11-29

**Authors:** Julia Orzelska, Amelia Trzcińska, Natalia Gierulska, Katarzyna Lachowska, Karolina Mazur, Rafał Tarkowski, Iwona Puzio, Ewa Tomaszewska, Anna Kułak, Krzysztof Kułak

**Affiliations:** 1Student’s Scientific Association, I Chair and Department of Gynaecological Oncology and Gynaecology, Medical University of Lublin, Staszica 16 Str., 20-081 Lublin, Poland; amelia.trzcinska33@gmail.com (A.T.); gierulskanatalia@icloud.com (N.G.); k.lachowska78@gmail.com (K.L.); k.mazur.l@wp.pl (K.M.); 2I Chair and Department of Gynaecological Oncology and Gynaecology, Medical University of Lublin, Staszica 16 Str., 20-081 Lublin, Poland; rafaltar@yahoo.com (R.T.); krzysztof.kulak@gmail.com (K.K.); 3Department of Animal Physiology, Faculty of Veterinary Medicine, University of Life Sciences, Akademicka 13, 20-950 Lublin, Poland; iwona.puzio@up.edu.pl (I.P.); ewarst@interia.pl (E.T.); 4Department of Diagnostic and Microsurgery of Glaucoma, Medical University of Lublin, 20-059 Lublin, Poland; puzioan@gmail.com

**Keywords:** antibody-drug conjugate, trastuzumab deruxtecan, ovarian cancer, targeted therapy

## Abstract

**Background/Objectives**: The treatment of ovarian cancer (OC), which is predominantly diagnosed in advanced stages, poses a significant challenge to modern gynecologic oncology practice. A significant proportion of patients exhibit chemoresistance, underscoring the need for novel therapeutic interventions. This challenge is further compounded by the immunogenic nature of this neoplasm, prompting the exploration of alternative therapies. A notable example is the use of trastuzumab-deruxtecan (T-DXd), an antibody-drug conjugate (ADC), that has demonstrated encouraging outcomes in preliminary studies and has the potential to become a new treatment option. This systematic review aims to prove that. **Methods**: The Preferred Reporting Items for Systematic Reviews and Meta-Analyses (PRISMA) structure was employed to systematically search the PubMed and Scopus databases from December 2024. Furthermore, authors employed materials from the FDA’s official website and registry of clinical trials that are currently recruiting participants for T-DXd’s studies. Eligible studies included randomized controlled trials and observational studies assessing T-DXd in patients with OC. Outcomes of interest were objective response rate (ORR), median overall survival, adverse effects, and progression-free survival. Data was synthesized narratively. **Results**: Following a thorough review of available literature, 30 scientific papers were selected for inclusion. A total of 598 patients participated in clinical trials. The most common adverse effects were blurred vision and nausea, generally manageable. The risk of bias was low in most studies. **Conclusions**: T-DXd shows promising efficacy. A comparison of T-DXd with the ADC currently approved for OC therapy reveals that both demonstrate similar median overall survival and ORRs. However, the drug has exhibited significant adverse effects in breast cancer trials and has been studied on a relatively small number of patients. Therefore, further clinical trials focusing on OC patients are necessary to better assess the safety and efficacy of T-DXd in this population.

## 1. Introduction

Ovarian cancer (OC) is undoubtedly one of the biggest challenges in modern gynecologic oncology. According to the latest data, 314,000 patients are diagnosed with it per year [1]. Around 80% of these cases present as advanced cancers with 5-year survival rates at 27% (stage III) and 13% (stage IV) [2], which makes OC the second most lethal gynecologic malignancy in Europe [3] and the eighth most fatal tumor found in women worldwide [1]. Due to such a poor prognosis, the mortality rates have only slightly declined in the past decades, in contrast with a more pronounced declining rate of morbidity. Researchers indicate this correlation has to do with an increase in the use of oral contraceptive pills (which are proven to lower the risk of OC), and a decrease in menopausal hormone therapy (which is proven to elevate the risk of OC). However, delays in diagnoses are still observed in addition to an inequality of access to health care [1].

The most common OC is a tubo-ovarian carcinoma (also known as an epithelial OC), which constitutes more than 90% of malignant cases, with a high-grade serous carcinoma as the most lethal subtype [2,3]. Less prevalent subtypes such as high-grade endometrioid carcinoma, low-grade serous carcinoma, and clear-cell carcinoma are also less fatal since they are usually diagnosed at earlier stages [3,4]. Their 5-year survival rates are, respectively: 82%, 71%, 66% [5].

There are several risk factors that can increase the morbidity of OC. Some of them are modifiable, since they are related to lifestyle, such as smoking, obesity, and inactivity. Studies indicate a link between them and less common histotypes, but there seems to be no correlation between lifestyle and the incidence of high-grade serous carcinoma [6]. It is also believed that adipocytes are a source of energy for OC cells, which allows them to grow and migrate. It also explains the association between OC and diabetes mellitus or metabolic syndrome [7].

Unmodifiable risk factors consist of genetic mutations of the Fanconi anemia pathway, such as BRCA1, BRCA2 (they increase a lifetime risk of OC up to approximately 40% and 18%), PALB2, ATM, RAD51C/D and BRIP1. Mismatched DNA repair pathway genes can cause an increased lifetime risk of OC, such as MLH1 (11%), MSH2 (17.4%), and MSH6 (10.8%) [8]. Studies show patients with breast cancer are twice as likely to develop a subsequent primary OC [9]. Another condition that can enhance the morbidity of OC is endometriosis, especially ovarian and deep infiltrating types. They have a 9.7-fold higher risk in comparison to women without endometriosis. Similarly to the impacts from an individual’s lifestyle, endometriosis affects a larger number of less common histotypes, such as endometrioid, low-grade serous, and clear-cell, rather than a high-grade serous carcinoma [10].

The standard treatment for patients with advanced tubo-ovarian carcinoma is a primary cytoreduction surgery with additional platinum-based chemotherapy. Women who could not qualify for the operation or in cases when complete cytoreduction to R0 (less than 1 cm) seems unlikely, the most sensible option is neoadjuvant chemotherapy [11]. According to observations presenting no clear survival benefit for introducing adjuvant chemotherapy to patients with stage IA or IB, only surgical treatment is possible to conduct. An adjuvant therapy (3 to 6 cycles) is recommended for stages IC2, IC3, and II. Most specialists agree that it can be advised for the treatment of stage IC1 tumors [3,11]. The most common combination chemotherapy consists of cisplatin and paclitaxel or carboplatin and paclitaxel [11]. However, there is a noticeable chemotherapeutic resistance which is especially troublesome among patients with recurrences—due to this, new substances and combination therapies are explored [11,12]. Research shows that in patients’ tumors, peripheral blood and ascites, non-spontaneous antitumor immune responses can be found, which makes OC an “immunogenic tumor”. Unsurprisingly, this knowledge was used to explore options for introducing immunotherapy to a variety of treatments. Combination therapies connect immunotherapy with chemotherapy, radiotherapy, anti-angiogenesis drugs, or PARP inhibitors. Each link improves the effectiveness of treatment in different patients based on their needs and unique responses [12].

A few years ago, antibody-drug conjugates (ADCs) became a breakthrough in oncology, which resulted in the approval of 12 substances by the Food and Drug Administration (FDA) in 2022. Since then, multiple clinical and preclinical trials have been run in order to prove the effectiveness of ADCs in different types of cancer and target more and more drug-resistant or poorly treated tumors. This group of substances is ideal for this purpose because they deliver cytotoxic payloads directly to cancer cells. ADCs consist of a monoclonal antibody that functions as a carrier for the potent chemotherapy agent [13]. The first ADC that was accepted to treat OC (more precisely, a platinum-resistant one) was mirvetuximab soravtansine-gynx (MIRV) [14]. Based on multiple articles, studies, and surveys, the authors of this study decided to explore the potential of using different ADCs in the treatment of OC. The authors believe that it is clinically meaningful to evaluate whether trastuzumab deruxtecan (T-DXd), although developed for a different molecular target, demonstrates outcomes that are comparable to, or potentially surpass, those achieved with MIRV. T-DXd specifically targets HER2-expressing cells, internalizes, and releases a topoisomerase I inhibitor payload via a cleavable linker. It leverages HER2 biology to expand therapeutic reach [15]. In this review, we discuss the mechanism of T-DXd recognizing OC, side effects of the drug, the application of this treatment, and its comparison to MIRV. By using the results from MIRASOL and SORAYA trials as reference points, the authors aim to assess how close T-DXd is to acceptance, thereby providing insight into its possible future role in the treatment of OC.

## 2. Materials and Methods

The search for this systematic review was conducted in accordance with PRISMA guidelines.

### 2.1. Search Method

A search for articles on T-DXd and OC was initiated in December 2024. It was conducted using the PubMed and Scopus databases. Authors supplemented the literature search data from the FDA’s official website and the clinical trials registry. These sources were reviewed to capture regulatory information, ongoing studies, and unpublished trial details relevant to ADCs in OC. No restrictions were put on the country or the data. The analysis was conducted exclusively on publications in English released after 2020, available in full text. The source materials were last reviewed on 25 March 2025. The following keywords were used as search terms: ‘trastuzumab’ AND ‘deruxtecan’ (PubMed—578 results; Scopus—1650 results), ‘trastuzumab-deruxtecan’ AND ‘adverse effects’ (PubMed—100 results; Scopus—50 results), ‘trastuzumab-deruxtecan’ AND ‘ovarian cancer’ (PubMed—18 results; Scopus—26 results), ‘mirvetuximab soravtansine-gynx’ AND ‘ovarian cancer’ (PubMed—7 results; Scopus—93 results).

### 2.2. Study Selection

The inclusion criteria contained publications from 2020 onward, written in English, addressing both human studies on women with OC as well as the in vitro activity of the drug in mouse models. The intervention method distinguished by the criteria was treatment with T-DXd, either alone or in combination with standard therapies. Additionally, the comparator included a placebo, standard-of-care treatment, or another active drug. At least one reported clinically relevant outcome was searched for, such as the ORR, median overall survival, adverse effects, and progression-free survival. In accordance with the established criteria, articles lacking credibility, duplicates, and those not written in English, as well as abstracts that are not related to the topic, were excluded from consideration. Of the articles found, 30 studies were chosen for review.

Five reviewers (J.O., A.T., N.G., K.L., K.M.) independently evaluated the titles and abstracts of the articles to identify those that met the inclusion criteria. The same five reviewers then assessed full texts of potentially relevant reports. Inconsistencies at either stage were resolved through discussion. When consensus could not be reached, three different reviewers (R.T., A.K., K.K.) were appointed to adjudicate. The number of records that were screened and the reasons for exclusion were documented in the PRISMA flow diagram (Figure 1).

## 3. Results

### 3.1. Fundamental Principles of the Pharmaceutical

T-DXd is a human epidermal growth factor receptor 2 (HER2)-targeted ADC. HER2 is responsible for cell proliferation, differentiation, and survival. This links its overexpression in tumors with poor prognoses, aggressiveness, a higher risk of recurrence, and a weaker response to chemotherapy. Therefore, there have been high hopes for new therapeutics such as T-DXd [15]. It has already been approved for the treatment of HER2-positive breast cancer, gastric cancers, non-small-cell lung cancer with an HER2 mutation, as well as being observedly beneficial in OC treatment [15,16].

T-DXd contains a humanized, anti-HER2 IgG1 monoclonal antibody, a tetrapeptide-based cleavable linker, and a topoisomerase I inhibitor payload, DXd [17]. The therapeutic connects with tumor cells through the HER2 receptor, where lysosomal enzymes (which are overexpressed in those cells) break it down and internalize it. This prevents downstream signaling through critical pathways like PI3K/AKT and RAS/MAPK, which are responsible for promoting cell division and inhibiting apoptosis [18]. The bond that connects the drug to the antibody makes it possible for the drug to be freed after internalization. Additionally, overexpression of lysosomal enzymes simplifies the release of the drug in tumor cells while simultaneously restricting the release of the substance in plasma [19]. This process results in the release of DXd that causes an inhibition of the topoisomerase-I-DNA complexes that inhibit replication of DNA, the cell cycle, and induce the apoptosis of tumor cells [15]. The high cell-membrane permeability of T-DXd enables a bystander effect that involves potent cytotoxic effects on carcinogenic cells adjacent to the primary target [19]. It is also worth noting that during the first approval, T-DXd achieved a higher drug-to-antibody ratio (≈8) than other approved ADCs with homogenous conjugation [17].

Dosage recommended by the FDA for adult patients with unresectable or metastatic HER2-positive solid tumors is 5.4 mg/kg given as an intravenous infusion once every 3 weeks (21-day cycle) until the disease progresses or there is an unacceptable amount of toxicity. Although parameters such as body weight, age, formulation, AST, and total bilirubin were proven to be statistically significant covariates, they also fail to constitute a clinically meaningful impact—this means there is no need to adjust dosages [19,20].

According to the primary results from the DESTINY-PanTumor02 Phase II Trial (which supports T-DXd’s potential in treating HER2-positive OCs), 85% of patients with OC experienced a drug-related adverse event. The most frequent ones were nausea, anemia, fatigue, decreased appetite, and diarrhea [15]. Pulmonary toxicity is one of the most severe adverse effects associated with concomitant T-DXd. It can manifest in various forms as pneumonia, pulmonary fibrosis, interstitial lung disease (ILD), or lung injury syndrome. The general clinical picture of toxicity is characterized by the presence of inflammation or scarring of the lung parenchyma, and emerging symptoms include mainly cough, occurring less frequently, dyspnea, fever, decreased exercise tolerance, cyanosis, pleural effusion, and pneumonia [16,21,22]. Key risk factors that result in an increased risk of developing lung function abnormalities during therapy include a prior medical history including lung disease, reduced lung function, smoking, Japanese or African American ethnicity, and ages older than 60 years [21]. ILD was found during the study with a frequency of up to about 15% of subjects. Most of it was mild, but death occurred in patients with severe cases (about 2%). For this reason, immediate discontinuation of treatment and the use of prednisolone to reduce inflammation are recommended immediately upon detection of ILD [22]. In addition to pulmonary toxicity, patients taking T-DXd should be wary of cardiac damage because of the drug. By blocking the HER2 receptor, it interferes with signaling pathways in the heart, leading to cardiomyocyte dysfunction. The entire mechanism promotes the development of heart failure, with the most common symptom being a decrease in left ventricular ejection fraction [23]. Moreover, the combination of trastuzumab and deruxtecan exacerbates oxidative stress in the heart and inhibits angiogenesis. An exacerbation of these symptoms can also occur in patients previously receiving anthracyclines and anthracycline chemotherapy. Symptoms that may indicate the onset of cardiac problems include shortness of breath, swelling, dizziness, as well as increased fatigue. If these occur and cardiotoxicity develops, the drug should be discontinued immediately, and it is recommended that cardiac function be monitored by echocardiography and laboratory tests that determine BNP [22,23]. In some cases, treatment of heart failure may include the use of drugs such as ACE inhibitors, beta-blockers, and diuretics to help improve cardiac function. However, this is only warranted if cardiotoxicity is confirmed by cardiac echo and BNP levels [23].

### 3.2. Outcomes of Clinical Trials

In order to even consider the use of T-DXd in the treatment of OC, it is beneficial to trace the path of MIRV approval and the differences between these drugs. Despite the fact that T-DXd and MIRV target different molecular antigens and have been investigated in distinct patient populations, their parallel consideration is clinically relevant. By invoking the concept of MIRV as a well-established ADC in OC treatment, the authors do not aim to suggest direct comparability, but rather to place T-DXd in the broader therapeutic context of ADCs. This perspective highlights the promise of ADCs as a class of drugs used in the treatment of OC. It also underlines the importance of future studies in order to determine the placement of T-DXd among new treatment strategies. MIRV targets folate receptor α (FRα), which is a marker that has high expression among the cancer cells and minimal expression across the healthy cells, minimizing the toxicity of the drug [24]. It connects the FRα with a specifically engineered monoclonal antibody, the antibody is linked to a cytotoxic agent called DM4. DM4 works by disrupting and destroying microtubules inside the cells, preventing mitosis and resulting in apoptosis of cancer cells [25]. A table was created to compare T-DXd with MIRV (Table 1).

Two trials were conducted—the SORAYA study and the MIRASOL study. The SORAYA study was a single-arm phase II study—the first to test the efficacy and safety of MIRV in adults with platinum-resistant OC. The study was conducted using 105 patients, all of whom had received bevacizumab prior; 51% had three prior lines of therapy, and 48% received a prior poly ADP-ribose polymerase inhibitor. Patients received single-agent MIRV with a dose of 6mg/kg given as an intravenous infusion once every 3 weeks [25,26]. The results confirmed that MIRV shows meaningful clinically antitumor activity and safety amongst patients with platinum-resistant OC, with an objective response rate (ORR) of 32.4% (5 patients with a complete response and 29 with a partial response) and a median duration of response being 6.9 months. 71.4% of patients experienced tumor reductions. The median overall survival was 15.0 months, and around 37% of participants were alive at year 2. Some treatment-related adverse effects were observed. The most common ones were blurred vision (41%), nausea (29%), and keratopathy (29%) [25,26]. The following study was the MIRASOL study, which was a confirmatory and randomized phase III clinical trial. It was designed to evaluate the effectiveness and safety of treating platinum-resistant, high-grade serous OC among adults with high FRα expression tumors, with MIRV in comparison to chemotherapy. This time, more patients underwent observation, with a population of 453 adult participants, all of whom had received one to three previous lines of standard antitumor treatment and documented progression despite therapeutics. Then, 227 patients were assigned to the MIRV group (they were receiving the same dosage as used in the SORAYA study) and 226 to the chemotherapy group (they were receiving paclitaxel, pegylated liposomal doxorubicin, or topotecan). Randomization was stratified according to the number of previous lines of therapy and chemotherapy agents. Once again, MIRV was proven to be the better clinical choice, as the ADC demonstrated a significant improvement in progression-free survival, which was 5.62 months in the MIRV group and 3.98 months with chemotherapy. Additionally, 12 patients, who were given MIRV, had a complete response and 84 had a partial response, while in the chemotherapy group none of the participants developed a complete response, and only 15 of them had a partial one. The ORRs were, respectively, 42.3% and 15.9% with a median overall survival of 16.46 months vs. 12.75 months. Adverse effects occurring in the MIRV group were similar to the SORAYA study, with 40.8% of patients experiencing blurred vision, 32.1% keratopathy, 30.3% abdominal pain, and 30.3% fatigue. MIRV’s effectiveness in delaying disease progression and ORR was notably higher, indicating greater tumor shrinkage in comparison with chemotherapy [24,27]. Both studies provided strong evidence for FDA approval and made MIRV one of the most important drugs available on the global market, as almost all of the patients suffering from recurrent OC will develop resistance to platinum. In the present time, only one study with T-DXd was conducted on patients with OC. The aforementioned DESTINY-PanTumor02 was an open-label, phase II study exploring the effectiveness of T-DXd (dosage of 5.4 mg/kg once every 3 weeks) among adults with HER2-expressing locally advanced, unresectable, or metastatic tumors with documented progression despite previous treatments or without alternative treatments. The trial had a sample size of 267, including 40 patients with OC [28]. In the immunohistochemical study, 15 patients with HER2 3+ OC and 25 patients with HER2 2+ OC were selected. Groups achieved an ORR of 42.5% (*n* = 17) according to the independent central review. This included a 66.7% ORR (*n* = 10) for HER2 3+ OC and a 28% ORR (*n* = 7) for HER2 2+ OC. The results demonstrated high efficacy, particularly in patients with high HER2 expression, who achieved an average of 13.9 months progression-free survival and 20.0 months median overall survival, in addition to a satisfactory response [15,28]. Most prevalent adverse reactions observed in this study were nausea (55%), anemia (37.5%), fatigue (27.5%), decreased appetite (20%), and diarrhea (20%). 52.5% of patients experienced grade 3 or higher drug-related adverse events, and 27.5% of patients experienced major adverse events. In 45% of cases, a dose adjustment was required [15]. A comparison of the results of those clinical trials, in conjunction with the established fact that MIRV is an approved drug for OC therapy, suggests that the results of the DESTINY-PanTumor02 trial may be considered promising. In light of these findings, it would be worthwhile to consider the implementation of large-scale clinical trials that exclusively enroll patients with OC, with the primary objective being the determination of HER2 expression in this specific patient population. It would not only confirm (or refute) the efficacy of T-DXd, but also allow for better verification of the safety of this drug and the relationship between HER2 expression and its action. A table was created to compare all aforementioned clinical trials (Table 2).

A cost analysis of T-DXd reveals that it generally exhibits a lower per-cycle cost compared to MIRV, although direct comparative cost-effectiveness studies in OC are lacking. It is important to note that if these trends are confirmed in future evaluations, this could accelerate the process of approval for T-DXd. However, the total lifetime treatment costs for both therapies can vary significantly based on individual patient factors and treatment duration [29,30].

## 4. Discussion

After analyzing the articles, one would have to ask: does T-DXd have the potential to become a newly approved therapy for OC? If so, what should be the focus in reaching this goal? OC continues to be a major challenge for gynecologic oncologists, ranking high on the list of most lethal cancers. By virtue of the fact that OC is most often detected when it is already at an advanced stage, and shows noticeable chemotherapy resistance and recurrence, standard treatments are inadequate in many cases [1,2,3,4]. In these situations, treating physicians often opt for combination therapy consisting of immunotherapy and chemotherapy, radiation therapy, anti-angiogenesis drugs, or PARP inhibitors [11,12]. What if these methods fail to achieve the desired outcomes?

Modern anti-cancer therapies focus on targeted drugs, which not only do an excellent job of nullifying specific cancer cells but additionally exhibit lower toxicity. One group of such drugs is ADCs, whose wave of popularity has also reached Gynecologic Oncology. T-DXd, discussed in this article, being an ADC, in theory, seems to be a satisfactory option for treating patients with HER2-positive OC [13]. This therapy, showing a bystander effect, first destroys tumor cells with receptors and then enters neighboring tumor cells lacking these specific receptors, destroying them as well. Thanks to this action, the development of cancer is stopped [19].

Although the number of studies treating OC with T-DXd is limited, the results that have been obtained and their comparison with the results of studies of already approved ADCs present a glimmer of hope for using T-DXd in the treatment of OC expressing HER2. Comparing the phase II trials on MIRV (the SORAYA study) and on T-DXd (the DESTINY-PanTumor02 study), we can see that both ORR and the median overall survival were higher among participants of the DESTINY-PanTumor02 study. It must be acknowledged that in the DESTINY-PanTumor02 study, the sample was quite small. Of the 267 patients enrolled in this study, only 40 had OC. Nevertheless, the encouraging signals of efficacy in this small group provide hope for patients and justify further investigation of T-DXd in larger, OC-specific trials. In addition, the number of ORRs varied significantly depending on the level of HER2 expression—10 patients with HER2 3+ and 7 patients with HER2 2+ achieved ORRs. This variable should also be taken into account in future studies and subjected to more in-depth analysis [15,25,26,27,28]. The severe side effects occurring after taking T-DXd should also be taken into consideration. Both pulmonary toxicity and cardiac damage can cause patient death, so doctors should closely monitor patients during T-DXd therapy and discontinue the drug as soon as the first symptoms of either appear. Risk factors should also be acknowledged, and serious consideration should be taken before beginning treatment in vulnerable patients [16,21,22,23].

At the time of writing this paper, T-DXd is being tested in two clinical studies. The DESTINY-PanTumor03 trial (NCT06271837) began in February 2024, and its primary completion is estimated for October 2025. It is a phase II trial. The expected number of participants is 175. However, the trial will contain patients with not only OC, but also with other solid tumors, so the number of cases of our interest may be unsatisfactory. Nevertheless, it will be another study to analyze [31]. A second study (NCT06819007) began in March 2025 with primary completion expected in November 2028. It is a phase III trial and is similar to the SORAYA study since it will also compare safety and efficacy between a group of patients receiving the study drug and a group receiving standard chemotherapy. The researchers designed the study for 582 people, so the number of participants is promising. Our belief is that the results will provide a lot of data, possibly allowing T-DXd to be approved for the treatment of OC [32].

## 5. Conclusions

A review of the current literature concluded that the immediate future of T-DXd therapy will be determined by the results of newly announced clinical trials. There is a belief that the data will coincide with that collected from the studies published so far, while maintaining a high safety profile for the drug. If it occurs, there will be ample evidence showing that T-DXd will be the next ADC approved for the treatment of OC. More targeted clinical trials would also be worth consideration in order to focus on a comparison of patient outcomes depending on the level of HER2 expression.

Given the challenges of treating patients with OC who do not respond to standard therapies, every new drug that comes to market offers more hope and improves patient survival. While a complete response may not be possible, due to the continuous and dynamic development of medicine, even a partial response is a huge success, as it increases the chances of prolonging patient lives until the next breakthrough therapy comes along. It is also worth noting that increased diversity in the pharmaceutical market can also increase the affordability of medicines.

This systematic review has several limitations related to the review process. Given that the present study exclusively considered works published in English, it may have introduced language bias and overlooked relevant findings in other languages. Exclusion of studies that provided not enough data on OC patients is also a potential limitation. Although this approach was intended to ensure the reliability of the findings, it is possible that it may have resulted in the omission of potentially relevant insights. However, efforts were made to minimize these risks by cross-checking data.

## Figures and Tables

**Figure 1 jcm-14-08483-f001:**
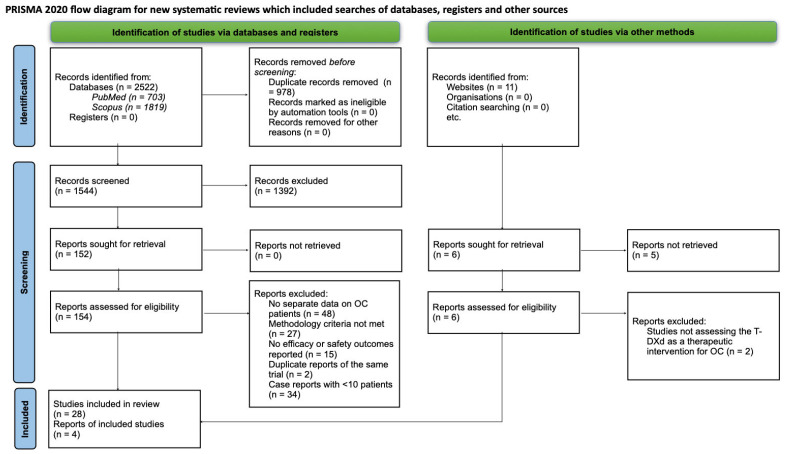
PRISMA flow diagram. Source: This work is licensed under CC BY 4.0. To view a copy of this license, visit https://creativecommons.org/licenses/by/4.0/ (accessed on 8 September 2025). The PRISMA checklist provided in Appendix A.

**Table 1 jcm-14-08483-t001:** Comparison between T-DXd and MIRV.

Characteristics	T-DXd	MIRV
target	HER2 (including HER2-low and HER2-mutant cases) [15,16,17,18,19,20,21,22,23]	FRα [24,25,26,27]
FDA approval for OC	not yet (in trials for HER2-positive OC) [15,16,17,18,19,20,21,22,23]	approved for FRα-positive, platinum-resistant OC in 2022 [24,25,26,27]
payload	deruxtecan: a topoisomerase I inhibitor disrupting DNA replication [15,16,17,18,19,20,21,22,23]	DM4: a maytansine derivative disrupting microtubule dynamics [24,25,26,27]
most severe adverse effects	pulmonary toxicity (including pneumonia, pulmonary fibrosis, ILD or lung injury syndrome), cardiac damage [15,16,17,18,19,20,21,22,23]	blurred vision [24,25,26,27]
trial examples	DESTINY-PanTumor02 [15,16,17,18,19,20,21,22,23]	SORAYA and MIRASOL [24,25,26,27]

**Table 2 jcm-14-08483-t002:** Comparison between completed studies performed on T-DXd and MIRV.

Characteristics	DESTINY-PanTumor02	SORAYA	MIRASOL
aim of the study	effectiveness of T-DXd among adults with HER-2 expressing locally advanced, unresectable, or metastatic tumors with documented progression despite previous treatments or without alternative treatments [15,28]	first to test the efficacy and safety of MIRV in adults with platinum-resistant OC [25,27]	evaluate effectiveness and safety of treating platinum-resistant, high-grade serous OC among adults with high FRα expression tumors, with MIRV in comparison to chemotherapy [26,27]
phase of the trial	II [15,28]	II [24,25,27]	III [24,26,27]
dosage of the drug	5.4 mg/kg given as an intravenous infusion once every 3 weeks [15,28]	6 mg/kg given as an intravenous infusion once every 3 weeks [24,25,27]	6 mg/kg given as an intravenous infusion once every 3 weeks [24,26,27]
number of participants	40 [15,28]	105 [24,25,27]	453 (227 in MIRV group, 226 chemotherapy group) [24,26,27]
ORR	42.5% [15,28]	32.4% [24,25,27]	42.3% in MIRV group (vs. 15.9% in chemotherapy group) [24,26,27]
median overall survival	20.0 months [15,28]	15.0 months [24,25,27]	16.46 months in MIRV group (vs. 12.75 months in chemotherapy group) [24,26,27]
most common adverse effects	nausea, anemia, fatigue, decreased appetite, diarrhea [15,28]	blurred vision, nausea, keratopathy [24,25,27]	blurred vision, keratopathy, abdominal pain, fatigue [24,26,27]

## Data Availability

The original contributions presented in this study are included in the article/Appendix A. Further inquiries can be directed to the corresponding author.

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
