# Peer review of "Feasibility of Trastuzumab-Deruxtecan in the Treatment of Ovarian Cancer: A Systematic Review"

_jcm, 2025, doi:10.3390/jcm14238483_

Round 1
Reviewer 1 Report
Comments and Suggestions for Authors
In this review, the authors highlight the feasibility of trastuzumab-deruxtecan (T-DXd) for ovarian cancer and compare it against MIRV. The topic is relevant, but several areas need strengthening:
-
The trial sections mainly re-summarize results without adding critical insights. Author interpretation would increase novelty and value.
-
A PRISMA flow diagram and more detail on the use of FDA and clinical trial registry materials would improve transparency.
-
Is there a cost-effectiveness factor that could eventually make T-DXd more attractive than MIRV? A brief discussion would help readers understand the clinical positioning.
-
The introduction is heavily weighted toward disease burden. A concise overview of ADC mechanisms, HER2 biology, and T-DXd’s design would provide a better balance.
Minor points:
-
Some sentences (e.g., lines 132–135) should be split for readability.
-
Correct spelling/formatting errors (e.g., lines 97, 173).
See above
Author Response
Thank you very much for taking the time to review this manuscript. Please find the detailed responses below and the corresponding revisions/corrections highlighted/in track changes in the re-submitted files.
Comments 1: The trial sections mainly re-summarize results without adding critical insights. Author interpretation would increase novelty and value.
Response 1: Thank you for this important comment. Our goal was to provide a comprehensive overview, but we agree that adding more author interpretation would strengthen the manuscript’s novelty and value. Changes can be found on page 7 in the revised manuscript: A comparison of the results of those clinical trials, in conjunction with the established fact that MIRV is an approved drug for OC therapy, suggests that the results of the DESTINY-PanTumor02 trial may be considered promising. In light of these findings, it would be worthwhile to consider the implementation of large-scale clinical trials that exclusively enroll patients with OC, with the primary objective being the determination of HER2 expression in this specific patient population. It would not only confirm (or refute) the efficacy of T-DXd, but also allow for better verification of the safety of this drug and the relationship between HER2 expression and its action. A table was created to compare all aforementioned clinical trials (Table 2).
Comments 2: A PRISMA flow diagram and more detail on the use of FDA and clinical trial registry materials would improve transparency.
Response 2: Thank you for pointing this out. We agree that including a PRISMA flow diagram and providing more detail on the use of FDA documents and clinical trial registry data would improve transparency. We added a PRISMA flow diagram (page 4) to illustrate the study selection process, and we expanded the Methods section (page 3) to clearly describe how FDA approval documents, clinical trial registry entries, and published literature were identified and integrated into the review. Mentioned changes can be found on page 3 and 4 in the revised manuscript: A search for articles on T-DXd and OC was initiated in December 2024. It was conducted using the PubMed and Scopus databases. Authors supplemented the literature search data from the FDA’s official website and clinical trails’ registry. These sources were reviewed to capture regulatory information, ongoing studies, and unpublished trial details relevant to ADCs in OC.
Comments 3: Is there a cost-effectiveness factor that could eventually make T-DXd more attractive than MIRV? A brief discussion would help readers understand the clinical positioning.
Response 3: Thank you for raising this important point. We agree that cost-effectiveness is a critical factor influencing the clinical adoption of novel therapies. Although direct cost-effectiveness analyses of T-DXd in ovarian cancer are not yet available, existing data from other tumor types suggest that the per-cycle cost of T-DXd is lower than that of MIRV. While these findings should be interpreted cautiously given the different trial populations and settings, the potential for T-DXd to offer a more cost-favorable profile is worth highlighting. We will add a brief discussion of this aspect to provide readers with a clearer sense of how cost considerations may shape the positioning of T-DXd in ovarian cancer treatment. Changes can be found on page 8 and 9 in the revised manuscript: A cost analysis of T-DXd reveals that it generally exhibits a lower per-cycle cost compared to MIRV, although direct comparative cost-effectiveness studies in OC are lacking.. It is important to note that if these trends are confirmed in future evaluations, this could accelerate the process of approval for T-DXd. However, the total lifetime treatment costs for both therapies can vary significantly based on individual patient factors and treatment duration [29, 30].; Given the challenges of treating patients with OC who do not respond to standard therapies, every new drug that comes to market offers more hope and improves patient survival. While a complete response may not be possible; however, due to the continuous and dynamic development of medicine, even a partial response is a huge success, as it increases the chances of prolonging patient lives until the next breakthrough therapy comes along. It is also worth noting that increased diversity in the pharmaceutical market can also increase the affordability of medicines.
Comments 4: The introduction is heavily weighted toward disease burden. A concise overview of ADC mechanisms, HER2 biology, and T-DXd’s design would provide a better balance.
Response 4: Thank you for this constructive suggestion. We agree that the introduction is currently weighted more towards disease burden, and that a concise overview of ADC mechanisms, HER2 biology, and the design of T-DXd would provide better balance and context for the reader. We revised the introduction and provided the changes that can be found on page 3 in the revised manuscript: Since then multiple clinical and preclinical trials were run in order to prove the effectiveness of ADCs in different types of cancer and target more and more drug-resistant or poorly treated tumors. This group of substances is ideal for this purpose because they deliver cytotoxic payloads directly to cancer cells. ADCs consist of a monoclonal antibody that functions as a carrier for the potent chemotherapy agent [13]. The first ADC that was accepted to treat OC (more precisely a platinum-resistant one) was mirvetuximab soravtansine-gynx (MIRV) [14]. Based off multiple articles, researches, and surveys; authors of this study decided to explore possibilities of using different ADC in the treatment of OC. Authors believe that it is clinically meaningful to evaluate whether trastuzumab deruxtecan (T-DXd), although developed for a different molecular target, demonstrates outcomes that are comparable to or potentially surpass those achieved with MIRV. T-DXd specifically targets HER2-expressing cells, internalizes, and releases a topoisomerase I inhibitor payload via a cleavable linker. It leverages HER2 biology to expand therapeutic reach [15]. In this review, we discuss the mechanism of T-DXd recognizing OC, side effects of the drug, application of this treatment and its comparison to MIRV.
Response to Comments on the Quality of English Language
Point 1: Minor points:
Some sentences (e.g., lines 132–135) should be split for readability.
Correct spelling/formatting errors (e.g., lines 97, 173).
Response 1: Thank you for your suggestions. All minor corrections, including improvements to readability, spelling, and formatting, have been implemented throughout the manuscript.

Reviewer 2 Report
Comments and Suggestions for Authors
Thank you for the opportunity to review this manuscript. The topic is clinically relevant, particularly given the urgent need for alternative therapies in platinum-resistant ovarian cancer. However, the review in its current form presents several conceptual and methodological concerns that should be addressed.
-
Lack of clear novelty: The manuscript does not clearly state what gap in current literature it aims to fill. While the therapeutic potential of T-DXd is discussed, the rationale for comparing it to MIRV—an approved ADC with a different molecular target and patient population—is not well justified.
-
Clinical relevance of the comparison: T-DXd targets HER2, while MIRV targets FRα. The populations included in their respective trials differ significantly. The manuscript risks "comparing apples to oranges" by juxtaposing these two agents without sufficient biological or clinical alignment. If the aim is to show that ADCs are promising in OC, this should be more explicitly framed.
-
Interpretation of results: The DESTINY-PanTumor02 trial included only 40 patients with OC. Drawing strong conclusions from such a limited dataset—especially when no phase III trials specific to OC are available—seems premature. The limitations of the available evidence should be acknowledged more clearly.
-
Recommendations for improvement:
-
Clarify the objective of the review and its intended contribution to the field.
-
Reassess the comparison between T-DXd and MIRV or provide stronger justification for it.
-
Include a more critical analysis of the evidence, especially regarding safety and trial design.
-
Explicitly discuss the limitations and avoid overstatements regarding efficacy.
-
This is a promising topic, but the manuscript would benefit from a more focused scope and a clearer, more critical narrative.
Comments on the Quality of English LanguageWhile the English is generally comprehensible, there are several awkward phrases and grammatical inconsistencies that affect the clarity of the text. A professional language edit is recommended.
Author Response
Thank you very much for taking the time to review this manuscript. Please find the detailed responses below and the corresponding revisions highlighted in the re-submitted files.
Comments 1: Lack of clear novelty: The manuscript does not clearly state what gap in current literature it aims to fill. While the therapeutic potential of T-DXd is discussed, the rationale for comparing it to MIRV—an approved ADC with a different molecular target and patient population—is not well justified.
Response 1: Thank you for pointing this out. The gap this manuscript aims to address is the lack of a comprehensive discussion of how emerging ADCs—particularly T-DXd—may fit into the treatment of ovarian cancer in comparison to the first approved ADC. We revised the manuscript to make it more explicit and to emphasize using MIRV as a benchmark on a way to acceptance of T-DXd to treat ovarian cancer. In the revised manuscript changes concerning this matter can be found on page 3: Based off multiple articles, researches, and surveys; authors of this study decided to explore possibilities of using different ADC in the treatment of OC. Authors believe that it is clinically meaningful to evaluate whether trastuzumab deruxtecan (T-DXd), although developed for a different molecular target, demonstrates outcomes that are comparable to or potentially surpass those achieved with MIRV. T-DXd specifically targets HER2-expressing cells, internalizes, and releases a topoisomerase I inhibitor payload via a cleavable linker. It leverages HER2 biology to expand therapeutic reach [15]. In this review, we discuss the mechanism of T-DXd recognizing OC, side effects of the drug, application of this treatment and its comparison to MIRV. By using the results from MIRASOL and SORAYA trials as reference points, authors aim to assess how close T-DXd is to acceptance, thereby providing insight into its possible future role in the treatment of OC.
Comments 2: Clinical relevance of the comparison: T-DXd targets HER2, while MIRV targets FRα. The populations included in their respective trials differ significantly. The manuscript risks "comparing apples to oranges" by juxtaposing these two agents without sufficient biological or clinical alignment. If the aim is to show that ADCs are promising in OC, this should be more explicitly framed.
Response 2: We agree the aim of the comparison needed more explicit phrasing. We fully acknowledge that T-DXd and MIRV target distinct antigens (HER2 vs. FRα) and that the populations enrolled in their respective pivotal trials differ substantially. Our intention was not to imply direct clinical interchangeability between these agents, but rather to place T-DXd within the broader context of ADCs in ovarian cancer and their acceptance path. In the revised manuscript changes concerning this matter can be found on page 5: In order to even consider the use of T-DXd in the treatment of OC it is beneficial to trace the path of MIRV approval and differences between those drugs. Despite the fact that T-DXd and MIRV target different molecular antigens and have been investigated in distinct patient populations, their parallel consideration is clinically relevant. By invoking the concept of MIRV as a well-established ADC in OC treatment, authors do not aim to suggest direct comparability, but rather to place T-DXd in the broader therapeutic context of ADCs. This perspective highlights the promise of ADCs as a class of drugs used in treatment of OC. It also underlines the importance of future studies in order to determine the placement of T-DXd among new treatment strategies.
Comments 3: Interpretation of results: The DESTINY-PanTumor02 trial included only 40 patients with OC. Drawing strong conclusions from such a limited dataset—especially when no phase III trials specific to OC are available—seems premature. The limitations of the available evidence should be acknowledged more clearly.
Response 3: Thank you for raising this important point. We appreciate this comment and agree that the ovarian cancer cohort in the DESTINY-PanTumor02 study (n=40) is small. Our intention was not to overstate the strength of the evidence, but rather to highlight the emerging signals of activity for T-DXd in ovarian cancer. We agree that without phase III data, conclusions must remain cautious, and we revised the manuscript to explicitly acknowledge these limitations. We clarified that current findings should be viewed as hypothesis-generating and underscore the importance of ongoing and future larger, ovarian cancer–specific trials. Changes can be found on page 8 in the revised manuscript: Although the number of studies treating OC with T-DXd is limited, the results that have been obtained and their comparison with the results of studies of already approved ADC present a glimmer of hope for using T-DXd in the treatment of OC expressing HER2. Comparing the phase II trials on MIRV (the SORAYA study) and on T-DXd (the DESTINY-PanTumor02 study), we can see that both ORR and the median overall survival were higher among participants of the DESTINY-PanTumor02 study. It must be acknowledged that in the DESTINY-PanTumor02 study the sample was quite small. Of the 267 patients enrolled in this study, only 40 had OC. Nevertheless, the encouraging signals of efficacy in this small group provide hope for patients and justify further investigation of T-DXd in larger, OC-specific trials. In addition, the number of ORRs varied significantly depending on the level of HER2 expression - 10 patients with HER2 3+ and 7 patients with HER2 2+ achieved ORRs. This variable should also be taken into account in future studies and subjected to more in-depth analysis [15,25-28]. The severe side effects occurring after taking T-DXd should also be taken into consideration. Both pulmonary toxicity and cardiac damage can cause patient death, so doctors should closely monitor patients during T-DXd therapy and discontinue the drug as soon as the first symptoms of either appear. Risk factors should also be acknowledged and serious consideration should be taken before beginning treatment in vulnerable patients [16,21-23].
Response to Comments on the Quality of English Language
Point 1: While the English is generally comprehensible, there are several awkward phrases and grammatical inconsistencies that affect the clarity of the text. A professional language edit is recommended.
Response 1: Thank you for this observation. We have carefully revised the manuscript to correct awkward phrasing and grammatical inconsistencies, and we sought the assistance of a native English speaker to ensure the text meets high standards of academic writing.

Round 2
Reviewer 1 Report
Comments and Suggestions for Authors
The authors have addressed all the comments.
Author Response
Comment 1: The authors have addressed all the comments.
Response 1: Thank you for your positive evaluation. We appreciate your time and are glad to hear that all comments have been satisfactorily addressed.
Reviewer 2 Report
Comments and Suggestions for Authors
I appreciate the thoughtful revisions made by the authors in response to the previous review. The manuscript now provides a more clearly articulated rationale for discussing trastuzumab deruxtecan (T-DXd) in the context of ovarian cancer, particularly in relation to MIRV. It is evident that the authors have taken care to clarify that the comparison is not based on direct clinical equivalence, but rather serves to situate T-DXd within the broader development of ADCs in this field.
The revised introduction offers a more focused background and highlights a relevant clinical question. The discussion of the DESTINY-PanTumor02 trial has been improved, with clearer acknowledgement of its limitations and a more cautious tone in interpreting its findings. This makes the overall argument more balanced and scientifically sound.
I do suggest, however, that the manuscript would benefit from a final round of English language editing. While the text is understandable and improved, there are still some awkward phrasings and grammatical issues that could be corrected to enhance clarity and professionalism.
For example, in the introduction (page 3), the sentence:
“Based off multiple articles, researches, and surveys; authors of this study decided to explore possibilities of using different ADC in the treatment of OC.”
contains several awkward constructions. A more appropriate academic formulation would be:
“Based on multiple articles, studies, and surveys, the authors of this study decided to explore the potential of using different antibody-drug conjugates (ADCs) in the treatment of ovarian cancer.”
I recommend a final round of professional English editing to address similar issues throughout the manuscript, which will enhance its readability and overall presentation.
Overall, the manuscript addresses a timely and relevant topic and presents a useful synthesis of the available evidence. With minor language improvements, I believe it will be a valuable contribution to the literature on emerging therapies in ovarian cancer.
Author Response
Comment 1:
I appreciate the thoughtful revisions made by the authors in response to the previous review. The manuscript now provides a more clearly articulated rationale for discussing trastuzumab deruxtecan (T-DXd) in the context of ovarian cancer, particularly in relation to MIRV. It is evident that the authors have taken care to clarify that the comparison is not based on direct clinical equivalence, but rather serves to situate T-DXd within the broader development of ADCs in this field.
The revised introduction offers a more focused background and highlights a relevant clinical question. The discussion of the DESTINY-PanTumor02 trial has been improved, with clearer acknowledgement of its limitations and a more cautious tone in interpreting its findings. This makes the overall argument more balanced and scientifically sound.
Response 1: We sincerely thank the reviewer for the thoughtful and constructive feedback. We are pleased that the revisions have improved the clarity, scientific balance, and contextual rational of our discussion on T-DXd in relation to MIRV and the broader development of ADCs in OC.
Comment 2: I do suggest, however, that the manuscript would benefit from a final round of English language editing. While the text is understandable and improved, there are still some awkward phrasings and grammatical issues that could be corrected to enhance clarity and professionalism.
For example, in the introduction (page 3), the sentence:
“Based off multiple articles, researches, and surveys; authors of this study decided to explore possibilities of using different ADC in the treatment of OC.”
contains several awkward constructions. A more appropriate academic formulation would be:
“Based on multiple articles, studies, and surveys, the authors of this study decided to explore the potential of using different antibody-drug conjugates (ADCs) in the treatment of ovarian cancer.”
I recommend a final round of professional English editing to address similar issues throughout the manuscript, which will enhance its readability and overall presentation.
Response 2: We appreciate the reviewer’s suggestion regarding English language. In response, we have completed an additional round of through language editing to address awkward phrasing, grammatical issues, and stylistic inconsistencies throughout the manuscript. The specific example noted by the reviewer has been corrected to the proposed, clearer form. We have made similar adjustments in all relevant sections to ensure improved readability and professionalism. All revisions made in response to this review have been marked in bold within the manuscript to ensure transparency. Regarding abbreviations, we chose to retain them throughout the manuscript to maintain consistency with common scientific writing conventions and to avoid excessive repetition of long medical terms. To support reader clarity, we have included a comprehensive list of abbreviations in an alphabetical order on the page 10 of manuscript.
Comment 3: Overall, the manuscript addresses a timely and relevant topic and presents a useful synthesis of the available evidence. With minor language improvements, I believe it will be a valuable contribution to the literature on emerging therapies in ovarian cancer.
Response 3: We thank the reviewer again for the constructive comments and positive response of the manuscript’s relevance and contribution. We believe that the additional language improvements, the clarification of abbreviations, and the clearly marked revisions further strengthen the clarity and overall quality of the text.